# Indoor Air Pollution, Related Human Diseases, and Recent Trends in the Control and Improvement of Indoor Air Quality

**DOI:** 10.3390/ijerph17082927

**Published:** 2020-04-23

**Authors:** Vinh Van Tran, Duckshin Park, Young-Chul Lee

**Affiliations:** 1Department of BioNano Technology, Gachon University, 1342 Seongnam-Daero, Sujeong-Gu, Seongnam-Si, Gyeonggi-do 13120, Korea; vanvinhkhmtk30@gmail.com; 2Institute of Research and Development, Duy Tan University, Da Nang 550000, Vietnam; 3Korea Railroad Research Institute (KRRI), 176 Cheoldobakmulkwan-ro, Uiwang-si 16105, Gyeonggi-do, Korea

**Keywords:** indoor air quality, indoor pollution, smart home, human diseases

## Abstract

Indoor air pollution (IAP) is a serious threat to human health, causing millions of deaths each year. A plethora of pollutants can result in IAP; therefore, it is very important to identify their main sources and concentrations and to devise strategies for the control and enhancement of indoor air quality (IAQ). Herein, we provide a critical review and evaluation of the major sources of major pollutant emissions, their health effects, and issues related to IAP-based illnesses, including sick building syndrome (SBS) and building-related illness (BRI). In addition, the strategies and approaches for control and reduction of pollutant concentrations are pointed out, and the recent trends in efforts to resolve and improve IAQ, with their respective advantages and potentials, are summarized. It is predicted that the development of novel materials for sensors, IAQ-monitoring systems, and smart homes is a promising strategy for control and enhancement of IAQ in the future.

## 1. Introduction

Indoor environment conditions contribute greatly to human wellbeing, as most people spend around 90% of their time indoors, mainly at home or in the workplace [1]. According to the World Health Organization (WHO), indoor air pollution (IAP) is responsible for the deaths of 3.8 million people annually [2]. IAP can be generated inside homes or buildings through occupants’ activities, such as cooking, smoking, use of electronic machines, use of consumer products, or emission from building materials. Harmful pollutants inside buildings include carbon monoxide (CO), volatile organic compounds (VOCs), particulate matter (PM), aerosol, biological pollutants, and others [3]. Therefore, over the past decade, research on air quality control has begun to shift from outdoor to indoor environments, as reflects lifestyle changes linked to increased levels of urbanization [4]. It was indicated that decreased IAQ can negatively affect human health by causing building-associated illness [5]. Both short- and long-term IAP exposure can cause a wide range of diseases [6]. Therefore, the development of monitoring systems has a vital role to play in IAQ control. 

IAP normally is a complex mixture of particulate and various gaseous components. IAP compositions differ significantly depending on sources, emission rates, and ventilation conditions [7]. For effective control of IAQ, therefore, it is necessary to determine the sources of air pollution. Moreover, the development of monitoring systems for the measurement of indoor pollutant concentrations as well as key strategies for control and enhancement of IAQ are considered essential. In this paper, we provide a comprehensive overview of the major IAP sources and IAQ-control strategies; we emphasize the sources, characteristics, and health effects of each IAP; we identify and discuss health issues and building-associated illnesses related to an IAQ decrease; and finally, we present the recent and trending strategies for the control and reduction of pollutant concentrations and better IAQ. It is expected that promising strategies for monitoring and control of IAQ in the future will include novel materials-based sensors, smart monitoring systems, and smart homes.

## 2. Indoor Air Quality (IAQ) and Indoor Air Pollution (IAP)

According to the EPA’s definition, IAQ is the air quality within and around buildings and structures, especially as it relates to the health and comfort of building occupants [8]. IAP, meanwhile, refers to the existence of pollutants, such as volatile organic compounds (VOCs), particulate matter (PM), inorganic compounds, physical chemicals, and biological factors, all of which are at high concentrations in the indoor air of non-industrial buildings, and all of which can have negative impacts on the human body. In order to protect people from such pollutants, IAQ has emerged and been developed as a research field [9]. The main parameters for evaluation of IAQ include pollutant concentrations, thermal conditions (temperature, airflow, relative humidity), light, and noise. Thermal conditions are crucial aspects of IAQ, for two basic reasons [10,11]: (i) Several problems related to poor IAQ can be solved simply by adjusting the relative humidity or temperature, and (ii) building materials in high-temperature buildings can be highly released. 

It has been indicated that IAQ in residential areas or buildings is significantly affected by three primary factors [12,13]: (i) Outdoor air quality, (ii) human activity in buildings, and (iii) building and construction materials, equipment, and furniture. It is known that outdoor contaminant concentrations and building airtightness have a great influence on IAQ, due to the possibility of transportation of contaminants from outdoors to indoors [14]. As outdoor pollutants’ concentrations increase, they are transported from outdoors to the indoor environment via ventilation. Hence, the correlation of outdoor air pollution with IAQ highly depends on the ventilation rate additionally to the lifetimes and mixing ratios of such pollutants [15]. Human daily activities generally cause IAP by the discharge of waste gases, tobacco smoke, pesticides, solvents, cleaning agents, particulates, dust, mold, fibers, and allergens [16]. Humans also create favorable conditions for the development of millions of mold, fungus, pollen, spores, bacteria, viruses, and insects, such as dust mites and roaches. Combustion sources and cooking activates contribute to carbon dioxide (CO_2_), sulfur dioxide (SO_2_), CO, nitrogen dioxide (NO_2_), and particulate matter (PM) emissions into indoor air environments [17,18]. In addition, equipment, such as computers, photocopy machines, printers, and other office machines, emit ozone (O3) and volatile compounds. Common building materials, such as poly(vinyl chloride) PVC floor covering, parquet, linoleum, rubber carpet, adhesive, lacquer, paint, sealant, and particle board, can shed toxic compounds (i.e., alkanes, aromatic compounds, 2-ethylhexanol, acetophenone, alkylated aromatic compounds, styrene, toluene, glycols, glycolesters, texanol, ketones, esters, siloxane, and formaldehyde) [10].

Importantly, the design and operation of ventilation systems also have a significant influence on IAQ. Due to superseding the stale indoor air by the fresh outdoor air, ventilation creates suitable IAQ and a healthy indoor environment. There are several benefits for the operation of ventilation in a building [19], including: (i) Providing oxygen and fresh air for human respiration; (ii) diluting indoor air pollutants to reach the short-term exposure limits of harmful contaminants as well as odors and vapors; (iii) using outdoor air with a low aerosol concentration to control aerosols inside buildings; (iv) controlling internal humidity; and (v) creating proper air distribution and promoting healthy and comfortable environment. Ventilation systems can be classified into two types [20], including: (i) Mechanical ventilation systems that use mechanical equipment, such as fans or blowers; and (ii) natural ventilation systems, which are the exchange processes between indoor air and out indoor without using mechanical equipment. Although natural ventilation systems may be well adopted by the occupants, they are insufficient in some buildings or climates. These days, mechanical ventilation systems have been commonly used in buildings, which significantly increases energy consumption. Thus, hybrid ventilation systems are designed to take advantage of both mechanical and natural ventilation systems, in order to decrease energy consumption and increase the use of sustainable technologies [21]. In hybrid ventilation systems, the shortcomings of natural ventilation will be compensated by mechanical components [22]. In summary, in heating, ventilation, and air conditioning (HVAC) systems of buildings, ventilation plays a key role in creating suitable IAQ, but it is also responsible for energy consumption. Therefore, improving ventilation systems in buildings is the key issue not only for enhancing energy efficiency but also for providing better IAQ to the occupants and minimizing the possibility of health problems as a consequence [21].

## 3. Main Pollutants in Indoor Air Environment

Numerous indoor air pollutants have been recognized to have harmful impacts on IAQ and human health [23]. The main indoor air pollutants include NOx, volatile and semi-volatile organic compounds (VOCs), SO_2_, O_3_, CO, PM, radon, toxic metals, and microorganisms. The sources and health effects of some common pollutants are listed in Table 1. Some of them can be present in both indoor and outdoor environments, while others originate from the outdoor environment. Generally, indoor air pollutants are able to be classified into organic, inorganic, biological, or radioactive [24]. 

### 3.1. Particulate Matters

PM is defined as carbonaceous particles in association with adsorbed organic chemicals and reactive metals. PM’s main components are sulfates, nitrates, endotoxin, polycyclic aromatic hydrocarbons, and heavy metals (iron, nickel, copper, zinc, and vanadium) [7]. Depending on the particle size, PM generally is classified into (i) coarse particles, PM_10_ of diameter <10 µm; (ii) fine particles, PM_2.5_ of diameter <2.5 µm; and (iii) ultrafine particles, PM_0.1_ of diameter <0.1 µm. PM is especially concerning, as it is sometimes inhalable, affecting the lungs and heart and causing serious health effects. It has been shown that indoor PM levels often exceed outdoor ones [25]. Indoor PM sources include (i) particles that migrate from the outdoor environment and (ii) particles generated by indoor activities. Cooking, fossil fuel combustion activities, smoking, machine operation, and residential hobbies are the main reasons why PM is distributed inside of buildings. Compared with PM_10_ and PM_2.5_, PM_0.1_ created by fossil fuel combustion represents a greater threat to health due to its penetrability into the small airways as well as alveoli [26,27]. According to research about the concentration of major indoor pollutants, it has been indicated that cooking and cigarette smoking are the largest sources of indoor air PM, whereas cleaning activities often have a lesser contribution to indoor PM [28]. Smoking is known as a major source of indoor PM_2.5,_ with estimated increases in homes with smokers ranging from 25 to 45 µg /m^3^ and the concentration in winter is greater than in summer [29]. For cooking activity, it was shown that cooking activities enable the emission of millions of particles (~10^6^ particles/cm^3^) through the burning of oil, wood, and food and most of them are ultra-fine particles [30,31]. In addition, these fine particles can distribute not only to the kitchen but also spread to the living room and other areas in the building, thereby causing adverse effects to the occupants’ health [31,32]. Meanwhile, other normal human activities, such as walking around and sitting on furniture, are likely to resuspend house dust and contribute to 25% of the indoor PM concentrations [29]. In summary, it has been found that the source strengths for human activities ranged from 0.03 to 0.5 mg.min^−1^ for PM_2.5_ and from 0.1 and 1.4 mg.min^−1^ for PM_10_ [33].

### 3.2. VOCs

Volatile organic compounds (VOCs) are recognized as gases containing a variety of chemicals emitted from liquids or solids [34]. Formaldehyde, a colorless gas with an acrid smell and which is released from many building materials, such as particleboard, plywood, and glues, is one of the most widespread VOCs. VOC concentrations in indoor environments are at least 10 times higher than outdoors, regardless of the building location [34,35]. Generally, indoor VOCs are generated from four main sources: (i) Human activities, including cooking, smoking, and the use of cleaning and personal care products; (ii) generation from indoor chemical reactions; (iii) penetration of outdoor air through infiltration and ventilation systems; and (iv) originating from building materials [35,36,37,38,39]. The VOCs concentration is able to be affected by air exchange rates, house age and size, building renovations, outdoor VOC levels, and door and window opening [40]. Moreover, it has been demonstrated that about 50 different VOCs are identified during 90-min cooking periods [35]. Because VOCs are organic chemicals that possess a low boiling point (T_b_) and are easily volatilized even at room temperature, the WHO classified them into four groups: (i) Very volatile organic compounds (VVOCs) with T_b_: 50–100 °C; (ii) volatile organic compounds (VOCs) with 100 °C < T_b_ < 240 °C; (iii) semi-volatile inorganic compounds (SVOCs) with 240 °C < T_b_ < 380 °C; and (iv) particulate organic matter (POM) with T_b_ > 380 °C [41,42]. Normally, exposure to VOCs released from consumer products is incurred via three main pathways: Inhalation, ingestion, or dermal contact. Most people are not seriously affected by short-term exposure to low concentrations of VOCs, but in cases of long-term exposure, some VOCs are considered to be harmful risks to human health, potentially causing cancer [43]. As for SVOCs, transdermal uptake directly from air has a higher contribution compared with intake via inhalation [44,45].

### 3.3. NO_X_

The two principal nitrogen oxides are nitric oxide (NO) and nitrogen dioxide (NO_2_), both of which are associated with combustion sources, such as cooking stoves and heaters [46]. Ambient concentrations of NO and NO_2_ vary widely depending on local sources and sinks. Their average concentration in buildings without combustion activities is half that in the outdoors, but when gas stoves and heaters are used, indoor levels often exceed outdoor levels. Under ambient conditions, NO is rapidly oxidized to form NO_2_; hence, NO_2_ is usually considered as a primary pollutant. The reaction of NO_2_ with water produces nitrous acid (HONO), a strong oxidant and common pollutant of indoor environments [47]. It has been indicated that indoor levels of NO_2_ are a function of both outdoor and indoor sources; hence, indoor levels can be influenced by high outdoor levels originating from combustion or local traffic sources. It was reported that the distance between buildings and roadways has a significant influence on indoor NO_2_ levels [48]. Besides, the air exchange between outdoors and indoors also affects NO_2_ levels in buildings [49]. Additionally, the major indoor sources include smoking and wood-, gas-, oil-, coal-, and kerosene-burning appliances, such as stoves, space, ovens, and water heaters and fireplaces [47].

### 3.4. Ozone 

Ozone is a powerful oxidizing agent mainly produced by photochemical reactions of O_2_, NOx, and VOCs in the atmosphere. However, it cannot be used to eliminate other indoor chemical pollutants, due to its slow reaction with most airborne pollutants [50,51]. Ozone enables rapid reaction with several indoor pollutants, but the reaction products can irritate humans and damage materials. The main sources of indoor ozone mainly come from the outdoor atmosphere and the operation of electrical devices [52]. The machines commonly emitting indoor ozone gas include photocopiers, disinfecting devices, air-purifying devices, and other office devices [53,54,55,56]. The ozone emission mechanisms of these devices can be divided into two categories: Corona discharge and photochemical mechanisms. It has been shown that indoor ozone levels depend on various factors: (i) The outdoor ozone level; (ii) indoor emission rates; (iii) air-exchange rates; (iv) surface-removal rates, and (v) reactions between other chemicals and ozone in the air [50]. Indoor ozone levels generally fluctuate between 20% and 80% of the outdoor ozone level according to the air-exchange rate [57]. Humans are exposed to ozone primarily by inhalation, but skin exposure is also a recognized vector [58]. 

### 3.5. SO_2_

Sulfur dioxide (SO_2_) is the most common gas among the group of sulfur oxides (SO_x_) present in the atmosphere. SO_2_ is primarily produced by the combustion process of fossil fuels, and combines with aerosols and PMs to form a complex group of distinct air [59]. Indoor sources of SO_2_ emissions include vented gas appliances, oil furnaces, tobacco smoke, kerosene heaters, and coal or wood stoves [60]. In addition, outdoor air is also regarded as a main source of indoor SO_2_ [61]. Indoor SO_2_ levels are often lower than outdoor levels. SO_2_ emission indoors is usually small, owing to its reactivation, which can be easily absorbed by indoor surfaces. It is known that the hourly concentration of SO_2_ in buildings is often below 20 ppb [62]. Human exposure to SO_2_, which can impair respiratory function, is only via inhalation. 

### 3.6. COx

Carbon monoxide (CO) in indoor air is produced mainly by combustion processes, such as cooking or heating. Besides, CO can also enter into indoor environments through infiltration from outdoor air [63]. The important sources of indoor CO emissions include unvented kerosene and gas space heaters; leaking chimneys and furnaces; back-drafting from furnaces, gas water heaters, wood stoves, and fireplaces; gas stoves; generators and other gasoline-powered equipment; and tobacco smoke [47]. The average concentration of CO in a building without any gas stoves is about 0.5–5 ppm, while the concentration in areas near gas stoves ranges from 5 to 15 ppm and even 30 ppm or higher. CO exposure can cause adverse health effects, such as (i) at low concentrations, there are impacts on cardiovascular and neurobehavioral processes; and (ii) at high concentrations, unconsciousness or death [64].

Carbon dioxide (CO_2_), a colorless and odorless gas, is a well-known constituent of the earth’s atmosphere and also a major human metabolite [65]. The average CO_2_ concentration in ambient air is about 400 ppm, which is primarily the result of the combustion of fossil fuels [65,66]. Recently, the indoor CO_2_ level has been applied as a reference for the assessment of IAQ as well as for ventilation control [66,67,68]. According to the ASHRAE standard, it is recommended that indoor CO_2_ concentrations are below 700 ppm to ensure human health [69]. It is established that exposure to a a CO_2_ concentration of 3000 ppm increases headache intensity, sleepiness, fatigue, and concentration difficulty [65,70].

### 3.7. Toxic Metals

Heavy metals are released into the atmosphere through either human activities or natural processes [71]. IAP by heavy metals has various causes, including infiltration of outdoor pollutants (dust and soil), smoking, fuel consumption products, and building materials [55]. Heavy metals in indoor dust, entering the human body through inhalation, ingestion, or dermal contact, can have adverse effects on human health [72,73]. According to the International Agency for Research on Cancer (IARC), heavy metals in indoor air are classified into two main groups based on their effects on humans: (i) Non-carcinogenic elements, including cobalt (Co), aluminum (Al), copper (Cu), nickel (Ni), iron (Fe), and zinc (Zn); and (ii) both carcinogenic and non-carcinogenic elements encompassing arsenic (As), chromium (Cr), cadmium (Cd), and lead (Pb) [74]. These common heavy metals (i.e., As, Cr, Cd, Pb) are likely to cause cancers [75,76], while Cd and Pb, along with some others, can cause carcinogenic effects, such as cardiovascular disease, slow growth development, and damage to the nervous system [73,77,78]. It has been reported that Pb levels in indoor air can fluctuate from 5.80 to 639.10 μg/g, while the highest levels of As, Al, Cr, Cd, Co, Cu, Ni, Fe, and Zn are about 486.80, 7150.00, 254.00, 8.48, 43.40, 513.00, 471.00, 4801.00, and 2293.56 μg/g, respectively [74].

### 3.8. Aerosols

Indoor aerosols are either primary aerosols originating from different indoor sources or secondary aerosols formed by indoor gas-to-particle reactions [79]. Moreover, outdoor particles infiltrating indoors are also likely to be a source of indoor aerosols. Secondary inorganic aerosols are PMs consisting of inorganic elements, including anthropogenic or crustal sources and water-soluble ions [6], while secondary organic aerosols (SOAs) are formed in the gas-to-particle conversion process of VOCs [80]. Additionally, carbonaceous aerosols, which comprise SOAs and elemental carbons released in incomplete combustion, are well-known species in PM_2.5_ [81]. Biological aerosols (bioaerosols) are a subset of atmospheric PMs comprising dispersal units (fungal spores and plant pollen), microorganisms (bacteria and archaea), or cellular materials [82]. Due to their diversity in terms of compounds and phases (gas, liquid, or solid), aerosols can be regarded as dynamic systems [83]. As such, their particle size distribution varies from the nucleation mode (<30 nm in vacuum cleaning condition) to the accumulation mode (~100 nm, indoor combustion aerosols from smoking, cooking, or incense burning), and to the fine and coarse modes (>1 µm, resuspension aerosols) [84,85]. Aerosol exposure through inhalation in the indoor environment has been linked to numerous adverse health effects, mainly in the lungs (the entrance to the human body) and other important target organs, such as the heart and brain [79].

### 3.9. Radon

The primary sources of indoor radon include building materials, soil gas, and tap water [86]. As soil contains radium at trace concentrations, radon is likely to be one of the constituents in the gas filling soil pores. As for radon emissions from building materials, all materials holding trace amounts of radium can release radon. Among building materials, masonry materials (i.e., stone, concrete, and brick) are the main sources for indoor radon emission, in that tons of such materials are used in building construction. Indoor radon can be released through the usage of water from underground water sources containing granite or other radium-bearing rock, and such water sources commonly contain radon concentrations above 10,000 pCi/L [87]. Finally, outdoor air is also regarded as a source of indoor radon [88]. Human exposure to radon in buildings is incurred mainly through the permeation pathways of underlying soil gas [89]. Epidemiological studies have demonstrated that indoor radon can cause lung cancer risk increases of 3% to 14%, depending on the average radon level [90].

### 3.10. Pesticides

These days, inorganic and organic pesticides have commonly been utilized as protectants for wooden building materials by impregnation or surface coating [91]. Pesticides are also used to control and prevent pests, including bacteria, fungi, insects, rodents, and other organisms [92,93]. In the indoor environment, pesticides are usually semi-volatile compounds that may exist in either gas or particulate form according to properties, such as the vapor pressure, product viscosity, and water solubility [94]. In addition, it has been indicated that carpet and textiles are likely to play the role of long-term reservoirs for organochlorine pesticides [95,96]. It is supposed that when used in carpets, textiles, and cushioned furniture, pesticides in fibers will migrate into polyurethane foam pads [97,98], and thus carpet, textiles, and cushioned furniture can reflect an integrated pesticide exposure during their lifetime. Moreover, pesticides are able to enter buildings from outdoors. Once inside, they can persist for months or years due to their protection against sunlight, extreme temperatures, rain, and other factors [97]. Dermal uptake, ingestion, and inhalation of particles or volatile compounds containing pesticides are believed to be potential exposure routes in the indoor environment [92]. Pesticide exposure is associated with adverse health risks, including (i) short-term skin and eye irritation, dizziness, headaches, and nausea; and (ii) long-term chronic impacts, such as cancer, asthma, and diabetes [99].

### 3.11. Biological Pollutants

Biological pollutants in indoor environments include biological allergens (e.g., animal dander and cat saliva, house dust, cockroaches, mites, and pollen) and microorganisms (viruses, fungi, and bacteria) [100]. Biological allergens, known as antigens, originate from a number of insects, animals, mites, plants, or fungi, and will induce an allergic state in reacting with specific immunoglobulin E (IgE) antibodies [101]. Indoor sources of allergens mainly include furred pets (dog and cat dander), house dust mites, molds, plants, cockroaches, and rodents [102], and there are outdoor sources as well [101]. Viruses and bacteria often originate from or are carried by people and animals. It has been demonstrated that exposure to biological allergens can result in sensitization, respiratory infections, respiratory allergic diseases, and wheezing [103], while exposure to bacteria and viruses indoors is likely to cause noninfectious and infectious adverse health outcomes [104].

## 4. IAQ Guidelines and Standards

It is obvious that the combination of long-term exposure and anthropogenic indoor activities can cause degradation of IAQ and significant risks to human health, even at low air pollutant concentrations. To face these IAQ problems, relevant organizations as well as the scientific community have attempted to develop and apply IAQ standards and guidelines. After many efforts, the world community established IAQ guidelines and standards based on an integrated building approach [110]. According to the WHO and USEPA, the role of IAQ guidelines is to provide a critical database as a reference for the prevention of harmful consequences of IAP and protection of public health; hence, the goal is to eliminate, or at least minimize, possible risks to human populations [111]. Table 2 summarizes the IAQ guidelines of the WHO and USEPA for some common pollutants [112]. However, it is necessary to distinguish non-occupational (i.e., residential houses, schools, offices) guidelines from occupational (industrial) standards [113]. Generally, the WHO and USEPA guidelines are the maximum concentrations during specific durations (i.e., 1 h, 24 h, or 1 year). Moreover, the IAQ guidelines of the WHO and USEPA seem non-applicable to occupational sectors [114], as they usually are applied for the control of IAQ inside households, schools, hospitals, public buildings, and offices [47]. In addition, each country will formulate specific standards or guidelines suitable to their own particular circumstances [115,116].

## 5. The Oxidative Capacity of Indoor Environment

The oxidative capacity has been significantly less explored in indoor environments. However, recent studies in atmospheric science have started looking at indoor oxidants and their precursors, in which the combination of indoor oxidants and their precursors will be referred to as “oxidants*” [117]. It has been supposed that ozone (O_3_), the hydroxyl radical (OH), and the nitrate radical (NO_3_) are major oxidants*** [118]. In addition, hydrogen peroxide (H_2_O_2_), nitrogen dioxide (NO_2_), hydroperoxy radicals (HO_2_), chlorine atoms (Cl), and alkylperoxy radicals (RO_2_) can be important indoor oxidants* under certain conditions. Oxidation is regarded as the dominant process in ozone reactions indoors; thus, most research into indoor oxidants* has focused solely on ozone. Recently, it was demonstrated that the formation of OH radicals by formaldehyde (HCHO) and photolysis of nitrous acid (HONO) is an important source of indoor oxidants* (Figure 1) [117]. In the atmosphere, the OH radical is known as the key species in photo-oxidation cycles; it can oxidize VOCs to form secondary aerosols or other gas species, which can cause toxic and carcinogenic effects in humans [119]. As for the important formation pathways of OH radicals, there are several different reactions productive of OH radicals, such as (i) NO and hydroperoxyl (HO_2_), (ii) ozone and alkenes, and (iii) the photolysis reaction of ozone (λ < 320 nm), HONO (λ < 400 nm), and H_2_O_2_ (λ < 360 nm) [120]. 

The primary sources of emissions of indoor oxidants* are building materials [121,122] and electronics [123]. Besides, acetaldehyde and other carbonyls can be emitted from human and microbial occupants [121,124,125], and moreover, human activities (i.e., cleaning or disinfection of surfaces, the use of air fresheners, cooking, smoking, vaping, bleach cleaning) are likely to be major sources of indoor oxidants* [117]. Other important sources are secondary oxidant formation indoors and transport from outdoors [50]. Greater outdoor-to-indoor transport of O_3_ and NO can lead to heightened NO/HO_2_ and O_3_/alkene reactions that result in higher levels of OH radicals.

## 6. The Effects of Indoor Air Pollution to Human Health

### 6.1. Building-Associated Illness

Over the past decades, various symptoms and illnesses have been linked to diminished IAQ in buildings and houses. Indoor exposure to inorganic, organic, physical, and biological contaminants, though often at low levels, is common, ubiquitous, and sustained. Therefore, the harmful effects of IAP on human health have always attracted great attention and concern. According to the WHO, building-associated illness refers to any illness caused by indoor environmental factors, which commonly are divided into two categories: Sick building syndrome (SBS) and building-related illness (BRI) [10]. Their associated symptoms are shown in Figure 2.

#### 6.1.1. Sick Building Syndrome (SBS)

SBS often refers to a group of symptoms that are linked to the physical environments of specific buildings [126]. Acute health and comfort effects of SBS will appear when patients spend a certain amount or duration of time in a building, but they and their causes are difficult to clearly identify [126]. These effects are either localized in particular areas or widespread throughout a building [127,128]. It has been reported that symptoms tend to worsen as a function of the exposure time in buildings and can disappear as people spend more time away from the building [129]. According to the WHO, SBS symptoms caused by IAP can be divided into four categories: (i) Mucous-membrane irritation: Eye, throat, and nose irritation; (ii) neurotoxic effects: Headaches, irritability, and fatigue; (iii) asthma and asthma-like symptoms: Chest tightness and wheezing; and (iv) skin irritation and dryness, gastrointestinal problems (i.e., diarrhea), and others [130,131]. The International Labour Organization (ILO) reported that infants, the elderly, persons with chronic disease, and most urban dwellers of any age have higher health risks linked with IAP-associated SBS symptoms [132]. A low ventilation rate, building dampness, and high room temperature also tend to increase the likelihood of SBS prevalence [133]. Moreover, other risk factors, such as gender, atopy, and psychosocial factors, also have a significant influence on SBS symptom prevalence [134].

#### 6.1.2. Building-Related Illness (BRI)

BRI describes illnesses and symptoms with an identified causative agent directly related to exposure to poor air quality in buildings. It is known that causative agents can be chemicals, such as formaldehyde, xylene, pesticides, and benzene, but biological agents are more widespread. In buildings, the typical sources for indoor emissions of biological contaminants are cooling towers, humidification systems, filters, drain pans, wet surfaces, and water-damaged building materials [10]. BRI symptoms have been associated with the flu, including fever, chills, chest tightness, muscle aches, and cough. In addition, serious lung and respiratory problems are likely to occur. Common BRI illnesses include Legionnaires’ disease, hypersensitivity pneumonitis, and humidifier fever [10]. It was reported that indoor environmental pollutants can cause BRI symptoms via four major mechanisms: (i) Immunologic, (ii) infectious, (iii) toxic, and (iv) irritant [135]. An irritant effect is often BRI’s initial insult, but toxic, allergic, or infectious mechanisms can arise subsequently, depending on the pollutant type and individual susceptibility. Psychologic mechanisms are often not paid significant attention but are demonstrably likely to increase the overall morbidity of building-related diseases as well [136].

Four main factors have been linked to BRI, including: (i) Physical-environmental factors, (ii) chemical factors, (iii) biological factors, and (iv) psychosocial factors. As for the physical-environmental factors, BRI can be influenced by the temperature, humidity, lighting, air movement, and dust concentration. The chemical factors, meanwhile, include various pollutants released from human activities and products, such as carpets, paint, new furniture, smoking, cosmetics, asbestos, drapes, and insecticides. Finally, major biological factors associated with BRI are microorganisms as mentioned in Section 3.11.

### 6.2. Acute Respiratory Infection

The respiratory system is frequently the primary target of IAP effects because pollutants often enter into the human body through inhalation. Depending on the area of the affected respiratory tract, acute respiratory infections can be classified into acute lower respiratory infections (ALRIs) and upper respiratory infections (URIs) [137]. URIs are illnesses involving the upper respiratory with common symptoms, such as cough, sinusitis, and otitis media [138], and they are often mild in nature and caused by biological pollutants (viruses, bacteria, fungi, fungal spores, and mites). Meanwhile, ALRI, an acute infection of the lung, is caused by viruses or bacteria, resulting in lung inflammation [137]. It has been found that IAP increases the risk of childhood ALRI by 78%, which leads to a million deaths in children under 5 years of age every year [139]. This may be because children have a relatively large lung surface area [140]. It has been demonstrated that children living in buildings using solid fuels possess a higher risk of developing ALRI by 2-3 times more than those combusting clean fuels [141]. IAP adversely affects specific and nonspecific host defenses of the respiratory tract against pathogens. In addition, IAP enhances the severity of respiratory infections and also results in high rates of chronic bronchitis of cooking mothers [142].

### 6.3. Pulmonary Diseases

Inhaled air pollutants are associated with allergic diseases and pulmonary diseases, such as asthma, atopic dermatitis, and allergic rhinitis. Moreover, smoking activity is regarded as one of the most important factors in the development of chronic inflammatory pulmonary diseases, including chronic obstructive pulmonary disease (COPD), asthma, and lung cancer [143]. Maximal obtainment of lung function can be significantly affected by IAP exposure; subsequently, lung function is declined. Noxious particles, including PM and CO, may influence lung development starting in utero [144].

COPD diseases are featured by an enhanced chronic inflammatory response in the airways and the lung to toxic PM or indoor air pollutants. It has been demonstrated that women, especially in developing countries, have a great risk for COPD because of exposure to household smoke from cooking [145]. PMs from fossil fuel combustion induce inflammation in the lung and highly reduce the pulmonary function. It has been indicated that fuel smoke in the household can cause COPD diseases with clinical signs and mortality similar to that of tobacco smokers [146]. Fuel burning generates chemical compounds with a high oxidative capacity that induce oxidative stress and DNA damage, which are pointed as the main mechanism responsible for the pathogenesis of COPD [147]. Additionally, biological factors, such as allergens, viruses, and bacterial substances, are likely to induce severe inflammatory reactions and cause immune dysfunction and chronic inflammation, which leads to COPD diseases [148,149,150]. 

Exposure to indoor air pollutants is able to result in asthma symptoms or cause asthma exacerbations [151]. It has been indicated that acute exposure to combustion smoke can induce bronchial irritation, inflammation, and enhance bronchial reactivity, which is regarded as the main mechanism responsible for asthma exacerbation, especially in children [152]. It has been demonstrated that children between 5 and 14 years living in houses combusting coal, wood, and kerosene have a relative risk of 1.6 in asthma exacerbation [153]. 

Lung cancer is usually associated with smoking. Recently, it has been demonstrated that lung cancer is highly linked to IAP exposure in females due to spending long times cooking [154] and thus lung cancer is more popular in females than males in non-smoking cases. Moreover, it has also been indicated that there is a considerable difference in the percentages of non-smoking females with lung cancer based on the various regions. For illustration, more than 80% of female lung cancer cases in east and south Asia are not related to smoking, whereas it is only 15% in the USA [155]. Emissions from the combustion of solid fuels for cooking or heating have been associated with a high risk of lung cancer [156]. It has been demonstrated that people using solid fuel, such as coal and wood, for heating and cooking throughout their life have a 4 times higher risk of lung cancer as compared with those combusting clean energy [156]. It is the incomplete combustion that highly emits various particle and gaseous carcinogens, including SO_2_, CO, NO_2_, PAHs, formaldehyde, heavy metals, and PM_2.5_ [157]. These pollutants have been found to be related to morbidity and mortality from respiratory diseases, especially lung cancer [158].

### 6.4. Cardiovascular Diseases (CVDs)

The use of solid fuels in households can emit various pollutants that are linked to CVDs, including PM, PAHs, CO, heavy metals, and other organic pollutants [159]. Exposure to PM_2.5_ increases the incidences of specific acute CVDs, such as ischemic stroke, myocardial infarction, cardiac arrhythmia, heart failure, and atrial fibrillation [160,161]. It has been reported that PM can cause CVDs due to inducing oxidative stress, systemic inflammation, increased blood coagulability, and autonomic and vascular imbalance [162]. PM is also a main factor causing significant increases in fibrinogen, platelet activation, plasma viscosity, and release of endothelins, a family of potent vasoconstrictor molecules. Moreover, CO in the indoor air environment is likely to influence tissue oxygenation through carboxyhemoglobin production, which results in a high impact on cardiovascular function [141]. Additionally, exposure to PAH and Pb from the use of fuel for cooking also enhances oxidative damage, stimulates the renin-angiotensin system, and downregulates nitric oxide [163]. These mechanisms may cause increased vascular tone and peripheral vascular resistance.

## 7. Current Strategies for Monitoring and Control of IAQ

### 7.1. Development of Materials for IAQ Sensors

During the past decade, two-dimensional (2-D) nanostructured materials, due to their unique physical and chemical properties, have attracted great attention for their utility in the design and production of gas-sensing devices. It was demonstrated that 2-D materials possess outstanding features, such as a large surface-to-volume ratio, excellent semiconducting properties, and high surface sensitivity. Moreover, the combination of 2-D nanostructured materials with other-dimensional materials is also proposed as a promising approach to the development of high-performance IAQ-monitoring sensors [164]. It has been demonstrated that 2-D nanostructured materials can be combined with 3-D bulk materials, 2-D nanostructures, 1-D nanostructures, and 0-D nanomaterials to produce 4 novel architectures for IAQ monitoring (Figure 3). Due to their special optical and electrochemical properties, 0-D nanomaterials of nanoscale (<100 nm) size, such as quantum dots (QDs), nanoparticles, nanocrystals, and nanoclusters, have been applied to the production of photonic, electronic, and chemical-sensing devices [165,166]. The use of 1-D nanostructures, such as nanotubes, nanowires (NWs), nanofibers, and nanorods, in combination with 2-D materials has been demonstrated to be effective in numerous IAQ-sensing applications, owing especially to their fast electron-transport capability, special morphology, and high surface-to-volume ratio [167]. Meanwhile, due to the formation of Van der Waals heterostructures through direct stacking of different 2-D materials, various types of 2-D–2-D heterostructures, including black phosphorus/MoSe_2_, graphene/MoS_2_, and graphene/WS_2_/graphene, have been applied in novel devices for IAQ monitoring [168,169]. Finally, by growing or transferring 2-D materials onto bulk 3-D semiconductors, layered 2-D-nanostructured semiconductors can be integrated with traditional bulk 3-D semiconductor substrates in the design of new gas-sensing devices [167]. In summary, heterostructure-based devices, given the superior properties of their constituent materials, could be utilized for a wide range of IAQ detection applications.

The development of inexpensive sensor networks and systems has emerged as a key strategy for the monitoring of IAQ. Metal oxide (MOx) sensors are currently regarded as one of several outstanding technologies, especially in consideration of their low cost, high sensitivity, simplicity, and compatibility with modern electronic devices [170]. With their small size and low cost, they are ideally suited for remote and portable monitoring systems. Recently, metal oxide semiconductor gas sensors have been considered as a viable platform for the detection of VOCs and CO in the indoor air environment [171,172,173]. It is reported that several metal oxides, including SnO_2_, ZnO, TiO_2_, In_2_O_3_, Fe_2_O_3_, MoO_3_, Co_3_O_4_, CuO, NiO, and CdO, have been utilized in IAQ-sensing applications.

### 7.2. Advanced Technologies for Monitoring of IAQ

There have been various techniques invented and introduced for real-time monitoring [174]. Among them, however, wireless sensor network (WSN) and Internet of Things (IoT)-based systems are the most popular technologies developed for IAQ monitoring due to their rising scope in the Industry 4.0 revolution.

#### 7.2.1. Internet of Thing (IoT)-Based Systems 

In recent years, with the development of mobile technologies, the Internet of Things (IoT), and big data, machine-learning technologies have been introduced as trending technologies that offer great capability for real-time IAQ monitoring. By the introduction of IoT-based portable IAQ-monitoring devices, these days, air quality can be easily monitored and controlled in real time. Many IoT-based IAQ-monitoring systems and devices have been designed and introduced, including open-source technologies for data processing and transmission [175].

One of the important applications of IoT in IAQ monitoring is “electronic noses” (E-noses), which are biomimetic-type devices that mimic the functionalities of mammals’ olfaction system [176]. Generally, an E-nose system consists of four basic components: (i) A multi-sensor array; (ii) software with digital pattern recognition algorithms; (iii) an information-processing unit (i.e., an artificial neural network (ANN)), and (iv) reference library databases [177]. By using the chemical sensor arrays in combination with the classification algorithms, E-noses can easily monitor target gases by detecting and discriminating types and concentrations [178]. Therefore, E-noses have globally attracted great interest for their low-cost portable IAQ-monitoring utility [179]. For example, Tastan et al. proposed a low-cost, portable, IoT-based, and real-time monitoring E-nose system [180], by which a range of sensors can measure different air pollutants (i.e., CO_2_, CO, PM10, and NO_2_) and air parameters (i.e., temperature and humidity). Especially, the proposed E-nose is produced with open-source, low-cost, and easy installation software and four detection units, including GP2Y1010AU (dust sensor), MH-Z14A (CO_2_ sensor), MICS-4514 (NO_2_ and CO sensor), and DHT22 (temperature sensor and humidity sensor). These sensors monitor via the 32-bit ESP32 Wi-Fi controller, while a mobile device interface based on the Blynk IoT platform receives data and records it in a cloud server (Figure 4A). This concept is anticipated to serve as a model for future IAQ-monitoring devices. In other research, Chen et al. developed a high-performance smart E-nose system comprising a multiplexed tin oxide (SnO_2_) nanotube sensor array, a read-out electronic circuit, a mobile phone receiver, a wireless data transmission unit, and a data processing application [181]. Compared with conventional devices, this nanotube sensor exhibited higher sensitivity gas detection and discrimination at room temperature, owing to the use of nanotube sensors possessing a large surface area for interaction with gas molecules. It was demonstrated that this E-nose device can detect indoor target air pollutants by only a simple vector-matching recognition algorithm; hence, it can be regarded as possessing state-of-the-art sensitivity for air pollutant detection at room temperature based on metal oxide sensors. Importantly, therefore, the smart E-nose device and system can be applied for high-performance monitoring of the indoor environment’s quality in smart buildings, smart homes, or even smart cities in the present and future (Figure 4B).

Recently, smart devices based on the integration of cloud computing and IoT have been developed to precisely monitor IAQ and efficiently transmit real-time data to a cloud computing-based web server using an IoT sensor network [182,183]. Jo et al. (2020) introduced a Smart-Air device to collect reliable and accurate data for IAQ monitoring (Figure 4C) [184]. The Smart-Air device includes three components: A pollutant detection sensor array, a microcontroller, and an LTE modem. The sensors in the Smart-Air device include a VOC sensor, a laser PM sensor, a CO_2_ sensor, a CO sensor, and a temperature/humidity sensor. The Smart-Air platform additionally relies on cloud computing technology and IoT technology to monitor and control IAQ anytime and anywhere. Importantly, the Korean Ministry of Environment testing has demonstrated that the device is highly reliable. All received data is stored in the web server’s cloud to provide resources for further analysis of IAQ [184].

According to the concept of the combination of ventilation system activation with IAQ-variable-level detection, a strategy for IAP reduction by demand-controlled ventilation systems (DCVs) based on CO_2_ detection was recently developed [185]. This system consists of four basic components, including the hardware, backend, mobile app, and IAQ index computation. The system operation depends on several real-time sensors and algorithms to accurately and automatically control the ventilation system and maintain a suitable concentration of air pollutants by balancing indoor and outdoor parameters [183]. As shown in Figure 4D, this sensing platform provides an IoT-based system for the control of IAQ levels in a building based on recorded data. Some parameters, such as the CO_2_, VOCs, humidity, and temperature of both the indoor and outdoor environments, are measured simultaneously; then, this data is applied for computation of the IAQ index, which is used for algorithm-based ventilation control. Additionally, the system allows for observation and management via a mobile app.

#### 7.2.2. Wireless Sensor Network (WSN)-Based Systems 

In the past few years, the wireless sensor network (WSN) has gained great attention in many different monitoring situations [186]. Generally, WSN is composed of tiny devices or nodes having an important role in collecting information by sensors from the environment as well as wirelessly communicating with other nodes in the system [174]. It has been indicated that in the development strategies of the WSN system, ZigBee, a wireless standard based on the IEEE 802.15.4 specification, is regarded as the most reliable communication protocol because of its low cost, low consumption, and low data rate [187,188]. In addition, use of the ZigBee protocol allows networks to work in power-saving mode [189,190].

In practical applications, WSN systems usually include a distributed sensor network connected to a cloud system [191]. ZigBee-based sensor nodes transmit field measurement data to the cloud system through a gateway. The optimized cloud computing systems have often been utilized to monitor, store, process, and visualize the obtained data from the sensor network [187]. Additionally, the obtained data is processed and analyzed by artificial intelligence techniques for optimization of the air pollutant detection. Due to the great number of nodes deployed to provide relevant information of the IAQ distribution in different areas, the WSN systems may significantly enhance the efficiency of air quality monitoring and measurement. The WSN is an automatic IAQ monitoring system with an autonomous, accurate, and simultaneous measurement of IAQ in different areas, which allows the user or the building manager to get data in real time [192]. It has been found that the WSN systems are usually applied for the monitoring and detection of numerous common VOCs, such as benzene, ethylbenzene, toluene, and xylene [191], and a variety of environmental parameters, including, temperature, relative humidity, CO, CO_2_, and luminosity [192].

### 7.3. Air Purification Technologies for IAQ Improvement

Control of emission sources, development of air purification technologies, and optimization of ventilation systems are regarded as three main approaches to improve IAQ [193]. Among them, the investigation of novel technologies to filter and purify indoor air pollutants has attracted great attention over the last decade. Adsorption technologies using carbon-based filter media have been known as the leading technologies for removing gas pollutants [194]. There are numerous residential and commercial HVAC filters that have been developed and used in various buildings, i.e., household, shopping mall, school, and office building. It was reported that adsorptive HVAC filters may effectively remove several noxious pollutants, including O_3_, aerosols, VOCs, and PMs [195]. The HVAC filters can be designed in various shapes, such as filter cassettes made of flat sheet filter media and cartridges inserted by granulated activated carbon or bag filters [196]. However, this technology has exhibited a significant limitation in terms of the adsorption efficiency because its equilibrium adsorption capacity is significantly decreased at sub-ppm concentrations of pollutants [197]. Additionally, it has been demonstrated that the adsorption efficiency of carbon-based filters is remarkably reduced in the presence of humidity. Therefore, improvement of the long-term operation and performance of this air filter system is a potential approach in the development strategies of HVAC filters.

Long-term operation can induce the proliferation of molds and the growth of germs, which lead to the effective decrease of the air filters [198]. Non-thermal plasma (NTP) technologies have been demonstrated as an effective solution to resolve these problems. The operation of NTP techniques is based on the generation of a quasi-neutral environment, such as radicals, ions, electrons, and UV photons [199]. Several non-thermal discharge plasma techniques have been developed for the removal of indoor air contaminants, such as NO_x_, SO_x_, and VOCs, including corona discharge [200], dielectric barrier discharge [201], and surface discharge. It has been shown that ozone formed by the electric discharge exhibits a significant germicidal power and the air filter can function as an effective electric precipitator due to electrostatic force. However, NTP technologies also have some disadvantages: (i) Poor energy efficiency; (ii) strong influence of humidity on the performance; and (iii) the formation of secondary hazardous pollutants, such as CO, HCHO, and NO_x_ [202]. These drawbacks have limited practical applications of this technique in the indoor air purification.

In the development strategies for indoor air pollution control, advanced oxidation processes (AOPs)-based technologies have attracted great attention due to the minimum generation of secondary pollution. Among them, photocatalysis, which is the process by which indoor air pollutants are disintegrated due to exposing semiconductor photocatalysts under sufficiently energetic irradiation, has been considered as a promising technology [203]. Compared with conventional adsorption technologies, photocatalysis exhibits several outstanding features: (i) Direct degradation of gaseous pollutants (particularly VOCs) into CO_2_ and H_2_O under ambient conditions; and (ii) applicability for the removal of low-concentration pollutants (sub-ppm levels) [204]. The semiconductors used as photocatalysts in the photocatalysis technology include titanium dioxide (TiO_2_), polymeric (or graphitic), tungsten oxide (WO_3_), carbon nitride (CN), Bi, Ag, cadmium sulfide (CdS), metal oxides MO_x_ (M = Fe, Zn, V), perovskite, and others [204]. However, due to the low cost, strong oxidative power, and chemical stability, TiO_2_ has gained more attention for IAP purification [205].

### 7.4. Smart Home for IAQ Control

In recent years, home automation generally and the “smart home” particularly have attracted great interest. Now, the smart home, controllable via smartphone applications and IoT-based wearable devices, is a vital element of smart cities. Its foundation was the proposed home energy management system (HEMS) [206,207]. In fact, originally, the smart home was developed simply to enhance the energy-usage efficiency [208,209] and optimize ventilation technology [21]. However, the term ‘smart home’ these days is most often associated with various home automation systems, such as home appliances, communications, and entertainment electronics. The future development of smart home technology is likely to proceed in two general directions: (i) The above-mentioned control of ventilation, blinds, and appliances for a reduction of energy costs and enhancement of health, safety, and comfortability; and (ii) assistance of elderly and disabled people by utilization of robotic machines and smart sensor technologies [210]. However, the smart home also offers opportunities to reduce the negative impacts of IAP on human health. Certainly, as smart home technology is increasingly applied in buildings, IAQ and air hygiene will be taken more seriously. Normally, a smart building system consists of three main components: (i) User gateway media, (ii) power supply media, and (iii) controller and devices [206]. User gateway media is a combination of a smartphone application and a supervisory control and data acquisition (SCADA) interface, which allows occupants to monitor and manage appliances remotely through smartphones or computers using Wi-Fi or a wireless module. Power supply media is a novel system that optimizes energy efficiency by using renewable energy sources. Finally, controller and devices provide automatic control of the fan speed, temperature, and air pollutant concentrations. The key to smart buildings is modern sensor technology [211]. By the use of multiple sensors, all climatic parameters and concentrations of polluting substances are recorded automatically and constantly in the SCADA system. Especially, based on the data recorded thereby, smart sensor systems can take appropriate action (e.g., adjustment of ventilation settings and/or temperature) to improve IAQ and reduce IAP [210].

## 8. Conclusions

In conclusion, pollutants in the indoor air environment are significant contributing causes of human diseases. There are numerous indoor air pollutants, including PM, VOCs, CO, CO2, ozone, radon, heavy metals, aerosols, pesticides, biological allergens, and microorganisms, all of which can lead to diminished IAQ and thereby harmful effects on human health. Most of these pollutants usually originate from two main sources: (i) Human activities in buildings, such as combustion, cleaning, use of certain building materials in the course of construction or renovation, and operation of electronic machines; and (ii) transportation from outdoor sources. Although these pollutants are often present at only low concentrations in buildings, long-term exposure can cause significant risks to human health. Generally, there are two categories of building-associated illness: Sick building syndrome (SBS) and building-related illness (BRI). To reduce IAP’s impacts, many strategies and approaches for the control and reduction of pollutant concentrations have been taken. It is expected that the development of advanced materials for sensors, IAQ-monitoring systems, and the smart home will prove to be effective for the control and enhancement of IAQ into the future.

## Figures and Tables

**Figure 1 ijerph-17-02927-f001:**
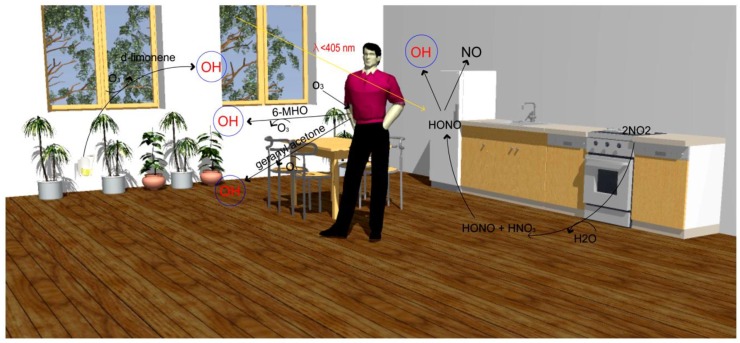
Schematic illustration of the important formation pathways of OH radicals indoors. Reproduced with permission from [118].

**Figure 2 ijerph-17-02927-f002:**
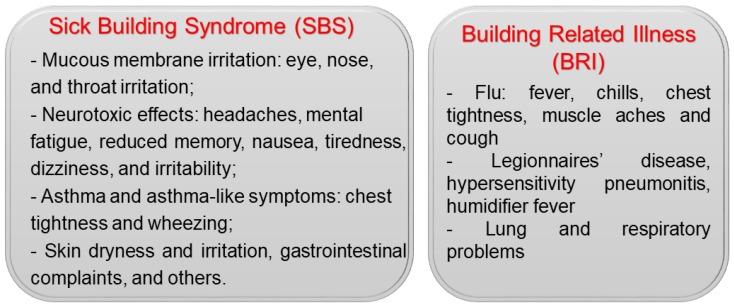
The common symptoms of sick building syndrome (SBS) and building related illness (BRI).

**Figure 3 ijerph-17-02927-f003:**
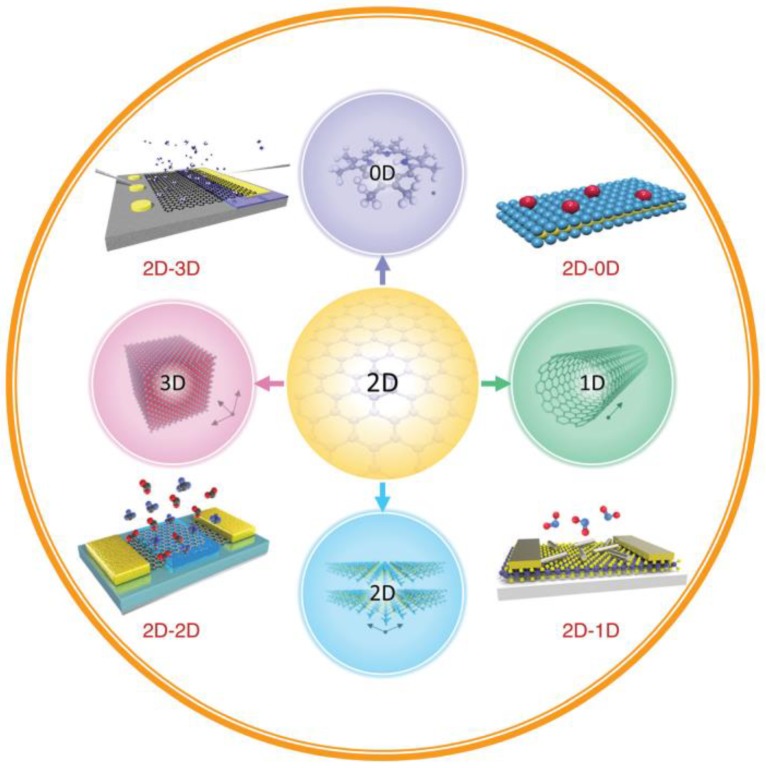
Schematic overview of 2-D nanostructured materials and different types of heterostructures with 0-D, 1-D, 2D, and 3-D materials used for gas-sensing applications. Reproduced with permission from [164].

**Figure 4 ijerph-17-02927-f004:**
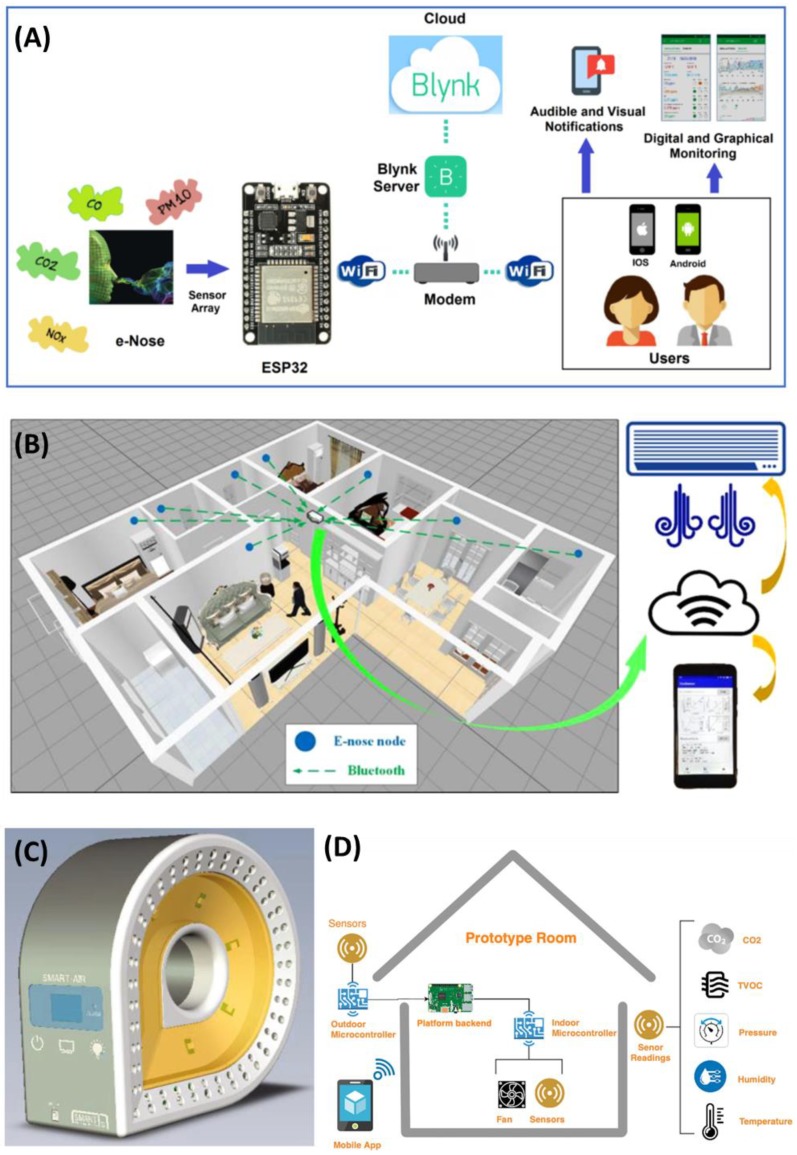
Illustration of the proposed electronics nose (E-nose) system architecture (**A**); the proposed E-nose application in future smart buildings (**B**); primitive concept design of Smart-Air (**C**); and a smart ventilation system for optimizing indoor air quality (IAQ) Levels in buildings (**D**). Reproduced with permission from [180,181,183,184].

**Table 1 ijerph-17-02927-t001:** Common indoor pollutants and their effects on human health.

Pollutants	Sources	Health Impacts	Refs
PM	Outdoor environment, cooking, combustion activities (burning of candles, use of fireplaces, heaters, stoves, fireplaces and chimneys, cigarette smoking), cleaning activities	Premature death in people with heart or lung disease, nonfatal heart attacks, irregular heartbeat, aggravated asthma, decreased lung function, increased respiratory symptoms	[7,25,26,27]
VOCs	Paints, stains, varnishes, solvents, pesticides, adhesives, wood preservatives, waxes, polishes, cleansers, lubricants, sealants, dyes, air fresheners, fuels, plastics, copy machines, printers, tobacco products, perfumes, dry-cleaned clothing, building materials and furnishings	- Eye, nose and throat irritation - Headaches, loss of coordination and nausea - Damage to liver, kidney and central nervous system - Some organics can cause cancer	[34,39,43,44,45]
NO_2_	Gas-fueled cooking and heating appliances	- Enhanced asthmatic reactions - Respiratory damage leading to respiratory symptoms	[46]
O_3_	Outdoor sources, photocopying, air purifying, disinfecting devices	DNA damage, lung damage, asthma, decreased respiratory functions	[51,52]
SO_2_	Cooking stoves; fireplaces; outdoor air	- Impairment of respiratory function - Asthma, chronic obstructive pulmonary disease (COPD), and cardiovascular diseases	[60]
CO_x_	Cooking stoves; tobacco smoking; fireplaces; generators and other gasoline powered equipment; outdoor air	Fatigue, chest pain, impaired vision, reduced brain function	[64,105]
Heavy metals	Pb, Cd, Zn, Cu, Cr, As, Ni, Hg, Mn, Fe Outdoor sources, fuel-consumption products, incense burning, smoking and building materials	- Cancers, brain damage - Mutagenic and carcinogenic effects: respiratory illnesses, cardiovascular deaths	[71,106,107]
Aerosols	Tobacco smoke, building materials, consumer products, incense burning, cleaning and cooking	Cardiovascular diseases, respiratory diseases, allergies, lung cancer, irritation and discomfort	[6,108,109]
Radon (Rn)	Soil gas, building materials, and tap water Outdoor air	Lung cancer	[86,89,90]
Pesticides	- Termiticides, insecticides, rodenticides, fungicides, disinfectants and herbicides - Building materials: carpet, textiles, and cushioned furniture - Outdoor environment	Irritation to eye, nose and throat; Damage to central nervous system and kidney; Increased risk of cancer	[92,93,97,98]
Biological allergens	House dust, pets, cockroaches, mold/dampness, pollens originating from animals, insects, mites, and plants	Asthma and allergies Respiratory infections, sensitization, respiratory allergic diseases and wheezing	[100,103]
Microorganism	Bacteria, viruses, and fungi are carried by people, animals, and soil and plants	Fever, digestive problems, infectious diseases, chronic respiratory illness	[100,104]

**Table 2 ijerph-17-02927-t002:** Indoor air quality guidelines for major indoor air pollutants.

Pollutants	Concentration Levels (mg/m^3^)	Exposure Time	Organization
CO	100	15 min	WHO
	60	30 min	
	30	1 h	
	10	8 h	
	29	1 h	USEPA
	10	8 h	
CO_2_	1800	1 h	WHO
NO_2_	0.4	1 h	WHO
	0.15	24 h	
	0.1	1 year	USEPA
PM	0.15	24 h	USEPA
	0.05	1 year	
O_3_	0.15–0.2	1 h	WHO
	0.1–0.12	8 h	
	0.235	1 h	USEPA
SO_2_	0.5	10 min	WHO
	0.35	1 h	
	0.365	24 h	USEPA
	0.08	1 year	
Pb	0.0005–0.001	1 year	WHO
	0.0015	3 months	USEPA
Xylene	8	24 h	WHO
Formaldehyde	0.1	30 min	WHO
Radon	100 Bq/m^3^	1 year	WHO

(Source: European Commission DG XVII: https://www.europeansources.info/corporate-author/european-commission-dg-xvii/).

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
