# Peer review of "Indoor Air Pollution, Related Human Diseases, and Recent Trends in the Control and Improvement of Indoor Air Quality"

_ijerph, 2020, doi:10.3390/ijerph17082927_

Round 1

Reviewer 1 Report

The authors present interesting information about indoor air quality and indoor air pollution. The authors reported the various forms of pollutants which impact the indoor air quality and later briefly describe the methods adopted for monitoring and controlling the indoor air quality. The manuscript in this form is not informative enough should not be considered for publication. As it is a review article, it is desired to provides process followed and the criteria in which the review work is carried out. Summarising various pollutants and their implication along with brief details about the new methods doesn’t add much to the existing literature. I recommend authors to refine the topic further to cover a specific part of the work in greater depth than covering the width of indoor air quality.

Author Response

Reviewer 1

The authors present interesting information about indoor air quality and indoor air pollution. The authors reported the various forms of pollutants which impact the indoor air quality and later briefly describe the methods adopted for monitoring and controlling the indoor air quality. The manuscript in this form is not informative enough should not be considered for publication. As it is a review article, it is desired to provides process followed and the criteria in which the review work is carried out. Summarising various pollutants and their implication along with brief details about the new methods doesn’t add much to the existing literature. I recommend authors to refine the topic further to cover a specific part of the work in greater depth than covering the width of indoor air quality.

Answer: Thank you so much for your comments. We added more detailed information and made some deep discussion. Briefly, we discussed the effect and importance of the ventilation system on indoor air quality. We also added and explained the human activities causing indoor air pollutants. Especially, we discussed some major health problems relating to indoor air pollution in detail. Finally, we added some advanced monitoring systems and air purification technologies for the control of indoor pollution. We hope that you can satisfy this current version of the manuscript. The supplementary contents marked in the green color in the manuscript.

Reviewer 2 Report

The paper by Tran et al. “Indoor air pollution, related human diseases, and recent trends in the control and improvement of indoor air quality” conducts a comprehensive review on what the indoor air pollution (IAP) is, sources, health effects, and available sensing and control strategies. I have a few comments below.
One big factor affecting indoor pollution is the ventilation rate (outdoor/indoor ratio). The authors need to expand on this. The author did, in a few places of the paper, compare indoor and outdoor concentrations. But not enough details to highlight the importance of ventilation.
For the description of pollutants and sources, the authors should discuss common indoor activities and resulting indoor air pollutants, like smoking and cooking usually introduced large amounts of pollutants to the indoor environment.
The effects of indoor air pollution on human health focused on two categories in the paper, sick building syndrome and building-related illness. Please consider discussing in terms of cardiovascular, lung cancer, acute lower respiratory infection, chronic inflammatory lung disease, etc.
The “control” part in the title seems to be missing in the main graphs. The author talked about smart home control, but very little information out there for how to reduce indoor air pollution. I assume discussions of filters are a good start.

L107 “VOC concentrations in indoor environments are at least ten times higher than outdoors, regardless of building location.”
Reference needed for this statement.

L155 “It is known that the hourly concentration of SO2 in buildings often is below 20 ppb.”
Reference needed for this statement.

L171 “400 ppm, which primarily is the result of the combustion of fossil fuels.”
Reference needed for this.

Author Response

Reviewer 2

The paper by Tran et al. “Indoor air pollution, related human diseases, and recent trends in the control and improvement of indoor air quality” conducts a comprehensive review on what the indoor air pollution (IAP) is, sources, health effects, and available sensing and control strategies. I have a few comments below.
1. One big factor affecting indoor pollution is the ventilation rate (outdoor/indoor ratio). The authors need to expand on this. The author did, in a few places of the paper, compare indoor and outdoor concentrations. But not enough details to highlight the importance of ventilation.

Thank you for your comments. We added the information (See line 83 – 103)

2. For the description of pollutants and sources, the authors should discuss common indoor activities and resulting indoor air pollutants, like smoking and cooking usually introduced large amounts of pollutants to the indoor environment.

Thank you so much for your comments. We added the information (See line 124-136, line 142-148, line 170-172)
3. The effects of indoor air pollution on human health focused on two categories in the paper, sick building syndrome and building-related illness. Please consider discussing in terms of cardiovascular, lung cancer, acute lower respiratory infection, chronic inflammatory lung disease, etc.
Thank you for your comments. We added the information (See line 384-line 449)
4. The “control” part in the title seems to be missing in the main graphs. The author talked about smart home control, but very little information out there for how to reduce indoor air pollution. I assume discussions of filters are a good start. Thank you so much for your comments. We added the information (See line 578-619)
5. L107 “VOC concentrations in indoor environments are at least ten times higher than outdoors, regardless of building location.” Reference needed for this statement.
Thank you for your comment. We added the reference (see line #142)
6. L155 “It is known that the hourly concentration of SO2 in buildings often is below 20 ppb.” Reference needed for this statement.
Thank you for your comment. We added the reference (see line #196)
7. L171 “400 ppm, which primarily is the result of the combustion of fossil fuels.”
Reference needed for this.
Thank you for your comment. We added the reference (see line #211)

Reviewer 3 Report

This manuscript discusses multiple aspects of indoor air quality - the causes, potential health impacts, and novel technology to measure it. It is overall well-written and assembled. There are some English language issues that should be corrected, but otherwise it is scientifically sound. My only major concern is Section 7.2, which I discuss later in the report). Other than that, the rest are minor suggestions.

Lines 78-80: The authors discuss important pollutants and these should include references.

Lines 94-95: The numbres following PM (i.e. 0.1, 2.5, 10) should be subscripted here (and throughout the manuscript) and should be listed as "ultrafine", "fine", and "coarse" respectively here rather than below (Lines 100-101).

Line 118: "As" should not be in blue color.

Line 121: The "2" in "NO2" should be subscripted.

Line 149: The "x" in "SOx" should be subscripted.

Line 159: The word "in" at the end of the line is not necessary.

Line 164: The word "The" should not be in blue.

Lines 174-175: Consider citing literature on this as there are many studies that have focused on this including educational and workplace related.

Lines 189-190: If available, it would be useful to list if there are guidelines for human health for these pollutants.

Line 198: Subscript "2.5" in "PM2.5".

Line 229: "and" should not be in blue.

Table 1: It would be more consistent to list the pollutants in the same order as they are described in Section 3 (i.e. PM, VOCs, etc.).

Line 258: "Table 2" should not be in blue.

Table 2: "3" in "mg/m3" should be superscripted.

Line 270 and onwards: It is unclear what "*" means. Could the authors clarify this please?

Line 276: "Figure 1" should not be in blue.

Figure 1: Some of the compounds are difficult to see due to the background. I recommend to set a frame or change the color scheme for greater visibility.

Line 299: "Figure 2" should not be bold.

Figure 2: Is this figure original from the authors? It seems to be of low resolution and grainy. The structure is confusing and could be better laid out. Currently it looks like BRI is a subset of SBS

Line 350: "Figure 3" should not be in blue.

Line 408: "Importantly too," is not necessary in this sentence.

Section 7.2 seems not well-structured. This is a small subset of available technologies and the authors need to acknowledge this more clearly. It would be useful to support the decisions to highlight these specific products otherwise it seems very ad hoc.

Author Response

Lines 78-80: The authors discuss important pollutants and these should include references.

Thank you for your comments. We inserted references (See line 82)

Lines 94-95: The numbres following PM (i.e. 0.1, 2.5, 10) should be subscripted here (and throughout the manuscript) and should be listed as "ultrafine", "fine", and "coarse" respectively here rather than below (Lines 100-101).

Thank you for your comments. We corrected it (See line 116-117 and line 122-123)

Line 118: "As" should not be in blue color.

Thank you for your comments. We corrected it (See line 156)

Line 121: The "2" in "NO2" should be subscribed.

Thank you for your comments. We corrected it (See line 159)

Line 149: The "x" in "SOx" should be subscribed.

Thank you for your comments. We corrected it (See line 189)

Line 159: The word "in" at the end of the line is not necessary.

Thank you for your comments. We corrected it (See line 199)

Line 164: The word "The" should not be in blue.

Thank you for your comments. We corrected it (See line 204)

Lines 174-175: Consider citing literature on this as there are many studies that have focused on this including educational and workplace related.

Thank you for your comments. We added more references (See line 216)

Lines 189-190: If available, it would be useful to list if there are guidelines for human health for these pollutants.

Thank you for your comments. There is no guideline for all heavy metals to human health. Currently, WHO and USEPA have only indoor air quality guidelines for Pb. Please see Table 2.

Line 198: Subscript "2.5" in "PM2.5".

Thank you for your comments. We corrected it (See line 239)

Line 229: "and" should not be in blue.

Thank you for your comments. We corrected it (See line 270)

Table 1: It would be more consistent to list the pollutants in the same order as they are described in Section 3 (i.e. PM, VOCs, etc.).

Thank you for your comments. We reordered it

Line 258: "Table 2" should not be in blue.

Thank you for your comments. We corrected it (See line 299)

Table 2: "3" in "mg/m3" should be superscripted.

Thank you for your comments. We corrected it (see Table 2)

Line 270 and onwards: It is unclear what "*" means. Could the authors clarify this please?

Thank you for your comments. We clarified it

Line 276: "Figure 1" should not be in blue.

Thank you for your comments. We corrected it (See line 319)

Figure 1: Some of the compounds are difficult to see due to the background. I recommend to set a frame or change the color scheme for greater visibility.

Thank you for your comments. This figure was cited from the ref [101] and we increased its contrast and brightness. We think that the current figure has a good visibility.

Line 299: "Figure 2" should not be bold.

Thank you for your comments. We corrected it (See line 343)

Figure 2: Is this figure original from the authors? It seems to be of low resolution and grainy. The structure is confusing and could be better laid out. Currently it looks like BRI is a subset of SBS

Thank you for your comments. We redraw it

Line 350: "Figure 3" should not be in blue.

Thank you for your comments. We corrected it (See line 460)

Line 408: "Importantly too," is not necessary in this sentence.

Thank you for your comments. We corrected it (See line 523)

Section 7.2 seems not well-structured. This is a small subset of available technologies and the authors need to acknowledge this more clearly. It would be useful to support the decisions to highlight these specific products otherwise it seems very ad hoc.

Thank you for your comments. We reorganized section 7.2. We hope you will satisfy with the current version.

Round 2

Reviewer 1 Report

After considering the new draft, I believe the paper is in the shape to be accepted. However, minor spelling check and grammar check is needed. 

Author Response

After considering the new draft, I believe the paper is in the shape to be accepted. However, minor spelling check and grammar check is needed. 

Answer: Thank you for your comment. We checked and corrected minor spelling and grammar errors.

The corrected contents are marked in green color. Please take a look again

Reviewer 2 Report

The authors addressed my comments.

Author Response

The authors addressed my comments.

Answer: Thank you for your comment. It makes the manuscript more perfectly